

# RT-Cabi: an Internet of Things based framework for anomaly behavior detection with data correction through edge collaboration and dynamic feature fusion

Xiaoshan Li and Mingming Chen

College of Information and Intelligent Mechatronics, Xiamen Huaxia University, Xiamen, China

## ABSTRACT

The rapid advancement of Internet of Things (IoT) technologies brings forth new security challenges, particularly in anomaly behavior detection in traffic flow. To address these challenges, this study introduces RT-Cabi (Real-Time Cyber-Intelligence Behavioral Anomaly Identifier), an innovative framework for IoT traffic anomaly detection that leverages edge computing to enhance the data processing and analysis capabilities, thereby improving the accuracy and efficiency of anomaly detection. RT-Cabi incorporates an adaptive edge collaboration mechanism, dynamic feature fusion and selection techniques, and optimized lightweight convolutional neural network (CNN) frameworks to address the limitations of traditional models in resource-constrained edge devices. Experiments conducted on two public datasets, Edge-IIoT and UNSW_NB15, demonstrate that RT-Cabi achieves a detection accuracy of 98.45% and 90.94%, respectively, significantly outperforming existing methods. These contributions not only validate the effectiveness of the RT-Cabi model in identifying anomalous behaviors in IoT traffic but also offer new perspectives and technological pathways for future research in IoT security.

# INTRODUCTION

## Background

The Internet of Things (IoT) is transforming the way we work and live. As the number of these devices increases rapidly, we are faced with unprecedented challenges in data processing and security. The vast amount of data generated by IoT devices requires not only real-time processing but also in-depth analysis to ensure the efficiency and security of the systems. The market value of IoT is expected to reach $534.3 billion by 2025, increasing the demand for real-time data monitoring. The projected number of IoT connections *via* LEO satellites is also on the rise—from six million in 2022 to 22 million by 2027, with an annual compound growth rate of 25%, highlighting the importance of real-time data processing (https://iot-analytics.com/number-connected-iot-devices/). The network security threats faced by IoT devices are significant, with data showing that on average,

Corresponding author
Mingming Chen,
xmumingming@163.com

each device is attacked within five minutes of connecting to the internet, and routers suffer an average of 5,200 attacks per month, underscoring the urgency of strengthening security measures (https://dataprot.net/statistics/iot-statistics/). The global number of IoT devices is expected to grow by 16%, reaching 16.7 billion by 2025 (https://iot-analytics.com/number-connected-iot-devices/). This reflects the scale of cross-industry integration and the ensuing data management challenges, highlighting the need for efficient processing solutions to address this trend.

As IoT technology continues to be widely applied, the security challenges it presents, particularly in identifying anomalous traffic behaviors, are becoming increasingly important (*Lee, Pak & Lee, 2020*). The diversity of IoT devices and the complexity of the data they generate make the patterns of abnormal behavior more varied and complex. Existing detection methods often struggle with this high-dimensional, complex data, finding it difficult to adapt and learn in a constantly changing environment (*Injadat et al., 2020*; *Di Mauro et al., 2021*). (1) The process of data collection often comes with errors and inconsistencies, leading to frequent occurrences of data loss or missing fields, not only increasing the difficulty of anomaly detection but also making the effective correction and completion of data an urgent problem to solve. (2) Considering the limited resources of IoT devices, such as processing power, storage space, and power, there is an urgent need for an efficient and energy-saving algorithm to address these challenges.

Therefore, facing the challenges of diversity in IoT devices, incompleteness of data, and limitations of device resources, traditional anomaly detection algorithms often fall short. This study introduced the RT-Cabi framework, which utilizes an adaptive edge collaboration mechanism, dynamic feature fusion and selection technology, and an optimized lightweight convolutional neural network (CNN) model. This approach not only improves data communication between sensors for better accuracy and completeness but also significantly lowers resource requirements.

## Literature review

### Current research on anomaly detection in IoT edge computing environments

In the context of IoT edge computing, the identification of anomalous behaviors is crucial for ensuring network security and the stable operation of devices (*Cui, Jiang & Xu, 2023*). With the explosive increase in the number of IoT devices and the diversification of application scenarios, traditional methods of anomaly detection face new challenges, particularly in dealing with novel network attacks, encrypted traffic analysis, and device heterogeneity (*Kamaraj, Dezfouli & Liu, 2019*; *Wijaya & Nakamura, 2023*; *Tong et al., 2023*).

These challenges have prompted innovative solutions. *Soukup, Čejka & Hynek (2019)* introduced a method for detecting behavioral anomalies by analyzing encrypted IoT traffic at the network edge, combining two semi-supervised techniques aimed at improving the reliability of anomaly detection and effectively mitigating the limitations of single techniques. However, it also noted that processing encrypted traffic requires more complex data analysis methods. *Kayan et al. (2021)* developed AnoML-IoT, an end-to-end data science pipeline that supports various wireless communication protocols and can be

deployed on edge, fog, and cloud platforms to address the challenges of IoT environment heterogeneity. Despite its promotion of anomaly detection mechanisms, its high requirements for multiple software tools and domain knowledge limit its widespread application. *Li et al. (2022)* proposed the ADRIoT framework, utilizing unsupervised learning with LSTM autoencoders and edge computing assistance, focusing on detecting network attacks in IoT infrastructures, especially unpredictable zero-day attacks. This method reduces reliance on labeled data and effectively improves the handling of new attack patterns, but it may limit the deployment and performance of detection modules on edge devices due to resource constraints.

### The potential of dynamic feature fusion and selection techniques in optimizing edge computing

Dynamic feature fusion and selection techniques, key to solving high-dimensional data problems and enhancing the processing capabilities of edge computing, have garnered widespread attention in recent years. Their potential application in optimizing edge computing is based on the latest research developments.

*Cai et al. (2018)* discussed feature selection methods that provide an effective pathway for high-dimensional data analysis, reducing computation time and improving the accuracy of learning models. Specific applications may require tailored feature selection methods. *Boulesnane & Meshoul (2018)* proposed a hybrid model that combines an online feature selection process with dynamic optimization, enhancing the quality of the selected feature set. However, the dynamic adjustment of the algorithm in practical applications requires fine-tuning according to the characteristics of the data flow. On the other hand, *Tubishat et al. (2020)* introduced an improved Butterfly Optimization Algorithm (DBOA) with a mutation-based local search algorithm (LSAM), effectively avoiding local optima, significantly improving classification accuracy, and reducing the number of selected features, which may require additional computational resources. *Wei et al. (2020)* presented an improved feature selection algorithm (M-DFIFS) by combining classical filters and dynamic feature importance (DFI), significantly enhancing performance within an acceptable computation time, although the algorithm has high complexity and sensitivity to parameters.

Dynamic feature fusion and selection techniques show significant potential for application in optimizing edge computing. Through refined algorithm design and efficient feature processing strategies, they can significantly improve the efficiency and accuracy of data processing in IoT edge computing environments.

### Research progress on adaptive collaborative frameworks and information sharing mechanisms

Research on adaptive collaborative frameworks and information sharing mechanisms is vital for enhancing system flexibility and efficiency. *Wang, Zheghan & Wu (2023)* proposed a content-aided IoT traffic anomaly detection approach that leverages both packet header and payload information to build machine learning models, achieving consistent detection results even under significant network condition changes. *Chatterjee*

*& Ahmed (2022)* conducted a comprehensive survey on IoT anomaly detection methods and applications, highlighting current challenges such as data and concept drifts and data augmentation with a lack of ground truth data. *Elsayed et al. (2023)* empirically studied anomaly detection for IoT networks using unsupervised learning algorithms, showing high F1-scores and area under curve (AUC) values with the novelty approach. *Eren, Okay & Ozdemir (2024)* reviewed XAI-based anomaly detection methods for IoT, providing insights into the transparency and interpretability of anomaly detection models. *Balega et al. (2024)* optimized IoT anomaly detection using machine learning models like XGBoost, support vector machine (SVM), and deep convolutional neural network (DCNN) demonstrating the superior performance of XGBoost in both accuracy and computational efficiency.

### The prospects of lightweight neural networks and multi-task learning in edge computing

The Edgent framework, proposed by *Li et al. (2019)*, facilitates collaborative inference of deep neural networks in a device-edge collaborative manner, particularly emphasizing the importance of DNN partitioning and appropriate resizing. It effectively reduces computational latency and enhances edge intelligence, though its adaptability to actual network fluctuations still needs further verification. Moreover, *Chen & Ran (2019)* delve into the challenges and solutions of applying deep learning in edge computing applications, offering perspectives on accelerating deep learning inference and distributed training on edge devices, despite the complexity and resource consumption of deep learning models remaining significant challenges.

Addressing the resource allocation problem in IoT networks, *Zhou et al. (2019)* discuss edge intelligence, emphasizing the integration of edge computing and artificial intelligence technologies to fully exploit the potential of edge big data. Challenges include system performance, network technologies, and management. *Liu, Yu & Gao (2020)* explored computational task offloading mechanisms through a multi-agent reinforcement learning framework, improving energy efficiency and reducing channel estimation costs, though its performance in highly dynamic environments requires further research. *Huang et al. (2022)* introduced a lightweight collaborative deep neural network (LcDNN) that significantly reduces model size and lowers mobile energy consumption by executing binarized neural network (BNN) branches on the edge cloud, demonstrating potential applications in mobile Web applications, though its performance and adaptability in complex tasks and variable environments need further evaluation.

In summary, the application prospects of lightweight neural networks and multi-task learning in edge computing are clear, providing strong technical support for real-time collaborative anomaly detection applications in IoT edge computing environments.

### Our contributions

This study identifies gaps in data integrity, algorithm adaptability, and computational resource optimization, detailed in Table 1. We introduce a comprehensive solution, the RT-Cabi framework, shown in Fig. 1. Figure 1 illustrates the integration of various

**Table 1 Literature review on anomalous behavior detection in IoT edge computing environments.**

| Author | Application scenario | Research content | Possible shortcomings |
|---|---|---|---|
| *Tubishat et al. (2020)* | Feature selection | Proposed DBOA avoids local optima effectively through LSAM | Requires additional computational resources |
| *Wei et al. (2020)* | Feature selection | M-DFIFS proposed combining filters and DFI to enhance performance | Algorithm complexity is high and sensitive to parameters |
| *Kayan et al. (2021)* | IoT environments | Developed AnoML-IoT supports various wireless communication protocols, deployable on edge, fog, and cloud platforms | High demands for multiple software tools and domain knowledge limit its widespread application |
| *Li et al. (2022)* | IoT infrastructure | Unsupervised learning method using LSTM autoencoder, focused on network attack detection | May limit the deployment and performance of edge device resource modules |
| *Wang et al. (2022)* | IoT traffic anomaly detection | Proposed content-aided approach leveraging packet header and payload information | May require more computational resources for processing payload data |
| *Chatterjee & Ahmed (2022)* | IoT anomaly detection | Survey on IoT anomaly detection methods and applications | Lack of comprehensive methods for integrating various sensors and data augmentation |
| *Elsayed et al. (2023)* | IoT networks | Empirical study using unsupervised learning algorithms for anomaly detection | Performance may vary with different datasets and network conditions |
| *Eren, Okay & Ozdemir (2024)* | IoT anomaly detection | Survey on XAI-based anomaly detection methods for IoT | Interpretability may come at the cost of reduced model complexity |
| *Balega et al. (2024)* | IoT security | Optimized anomaly detection using machine learning models like XGBoost, SVM, and DCNN | The approach's effectiveness may depend on the diversity of datasets and IoT environments |

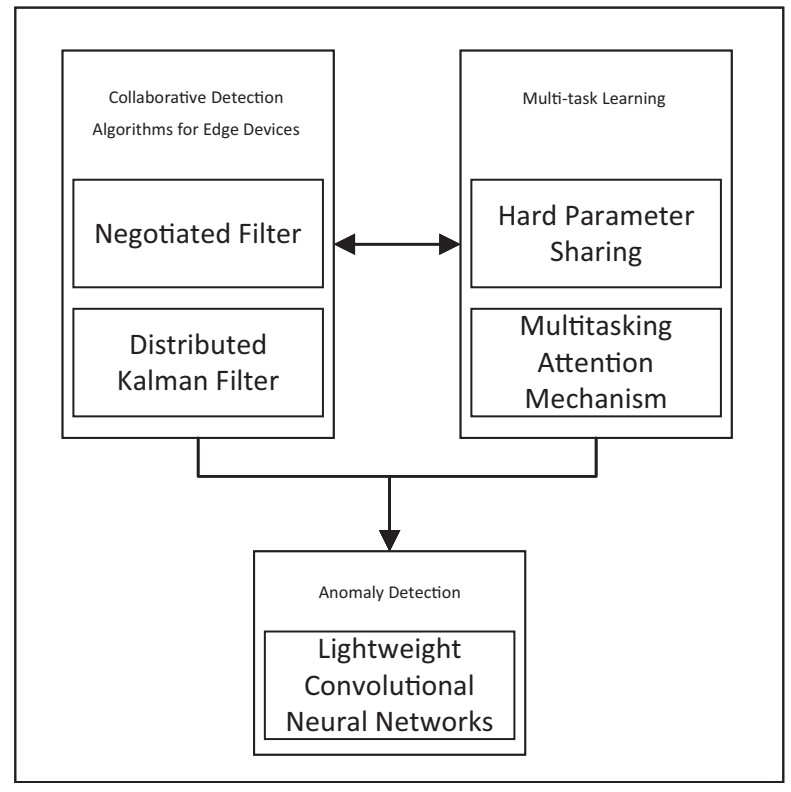

**Figure 1 The RT-Cabi framework.**

components, including negotiated filtering, distributed Kalman filtering, hard parameter sharing, multitasking attention mechanisms, and lightweight convolutional neural networks. These elements work together to enhance anomaly detection and collaborative detection algorithms for edge devices. The key contributions of our proposed framework are summarized as follows:

- **Adaptive anomaly detection for IoT:** We designed an edge collaborative framework based on adaptive parameter adjustment. This framework can capture environmental changes in real-time, dynamically adjust model parameters through weighted collaborative filtering and distributed Kalman filtering techniques. This approach ensures the model remains robust and responsive to new data patterns and anomalies.
- **Data anomaly correction and imputation strategy:** We developed a dynamic feature fusion and selection mechanism combining hard parameter sharing and multi-task learning technologies. By introducing adaptive weight adjustment and an advanced multi-task dynamic attention mechanism, this strategy can effectively handle common feature missing issues in the IoT environment. This ensures data integrity and enhances the overall reliability of the system.
- **Mathematical collaborative optimization strategy:** This study also proposes a set of mathematical collaborative optimization strategies, integrating methods from edge collaboration, feature fusion, and lightweight CNN optimization, forming a comprehensive model optimization scheme. This strategy reduces the computational demands and resource consumption in resource-constrained edge computing environments, making it practical for real-world IoT applications.

## THE RT-CABI FRAMEWORK

### IoT anomaly detection model

Consider an IoT environment composed of $N$ devices, denoted as $\mathcal{N} \overset{\Delta}{=} \{1, \ldots, N\}$. Each device $i \in \mathcal{N}$ can collect and process a local dataset $\mathcal{D}_i$, containing $D_i$ samples $\{x_i^n\}_{n=1}^{D_i}$. These samples are independently and identically distributed (i.i.d.) drawn from the local distribution $\mathcal{D}_i$ of device $i$, with each sample $x_i^n$ including a training input and its corresponding label. Assume the data across devices is heterogeneous, *i.e.*, the local distributions $\{\mathcal{D}_i\}$ are non-i.i.d.

Each device trains a local model composed of $d$ parameters, represented by the vector $\theta \in \Theta \subset \mathbb{R}^d$, using its dataset. The training objective is to minimize the local objective function $f_i(\theta)$ based on the loss metric $l(\cdot; \theta)$, with device $i$'s local objective defined as:

$$f_i(\theta) = \frac{1}{D_i} \sum_{n=1}^{D_i} l(x_i^n; \theta). \tag{1}$$

Thus, the goal of device $i$ is to find the parameters $\theta_i^*$ that minimize Eq. (1):

$$\theta_i^* = \arg\min_\theta f_i(\theta). \tag{2}$$

The server aims to train a global model with parameters $\theta$ using the data available from the user side. The global learning objective is given by the following equation:

$$F(\theta) = \frac{1}{N}\sum_{i=1}^{N} f_i(\theta). \tag{3}$$

Therefore, the server seeks to solve the following minimization problem:

$$\theta^* = \arg\min_\theta F(\theta). \tag{4}$$

We introduce adaptive parameters $\phi$, allowing the model to dynamically adjust according to changes in the environment, to enhance the accuracy and adaptability of anomaly detection in the IoT environment. The problem is transformed into a multi-objective optimization issue of simultaneously optimizing $\theta$ and $\phi$ to achieve optimal anomaly detection performance.

## Motivation for adaptive edge collaboration framework: achieving efficient information sharing and task distribution

- Existing edge computing frameworks often use centralized management or information exchange mechanisms based on simple protocols, which struggle in handling dynamically changing network environments and diverse device capabilities (*Hu & Huang, 2022*). This limitation leads to inefficient information sharing, inability to flexibly allocate tasks, and severely restricts the system's response speed and adaptability to change (*Wang et al., 2022*; *Patsias et al., 2023*).
- We propose an adaptive edge collaboration framework that integrates advanced algorithmic design, combining weighted collaborative filtering with distributed Kalman filtering. Its core innovation is the dynamic adjustment of task and resource allocation strategies in response to real-time network conditions and device capabilities, ensuring efficient resource use and quick task response under diverse conditions.

## Mathematical model of the adaptive edge collaboration framework

We focus on how the adaptive edge collaboration framework enables effective information sharing and task allocation among multiple devices in an edge computing environment. The state at time $t$ is $x_t$, following the dynamic model:

$$x_{t+1} = Ax_t + Bu_t + w_t \tag{5}$$

where $A$ is the state transition matrix, $B$ is the control matrix, $u_t$ is the control input at time $t$, and $w_t$ is the process noise, assumed $w_t \sim \mathcal{N}(0, Q)$, with $Q$ as the covariance matrix of the process noise.

The device observation at time $t$, $z_t$, is:

$$z_t = Hx_t + v_t \tag{6}$$

where $H$ is the observation matrix, and $v_t$ is the observation noise, assumed $v_t \sim \mathcal{N}(0, R)$, with $R$ as the covariance matrix of the observation noise. Estimation accuracy improves by merging information from different devices, described by the collaborative filtering algorithm:

$$\hat{x}_{t|t} = \frac{\sum\limits_{i=1}^{N} K_t^i z_t^i}{N} \tag{7}$$

where $K_t^i$ is the Kalman gain of device $i$ at time $t$, and $N$ is the total number of devices. Information fusion among devices uses a weighted collaborative filtering algorithm to enhance overall state estimation accuracy:

$$\hat{x}_{t|t} = \left( \sum_{i=1}^{N} \frac{\exp\left(-\frac{1}{2}||z_t^i - H\hat{x}_{t|t-1}^i||^2_{(R+Q)^{-1}}\right)}{\sum\limits_{k=1}^{N} \exp\left(-\frac{1}{2}||z_t^k - H\hat{x}_{t|t-1}^k||^2_{(R+Q)^{-1}}\right)} K_t^i z_t^i \right.$$
$$\left. + \sum_{j=1}^{M} \frac{\exp\left(-\frac{1}{2}||y_t^j - G\hat{x}_{t|t-1}^j||^2_{(S+P)^{-1}}\right)}{\sum\limits_{l=1}^{M} \exp\left(-\frac{1}{2}||y_t^l - G\hat{x}_{t|t-1}^l||^2_{(S+P)^{-1}}\right)} L_t^j y_t^j \right) / (N+M) \tag{8}$$

where $K_t^i = P_{t|t-1}H^T(HP_{t|t-1}H^T + R)^{-1}$ and $L_t^j = P_{t|t-1}G^T(GP_{t|t-1}G^T + S)^{-1}$ are the Kalman gains of device $i$ and external information source $j$ at time $t$. Observations $z_t^i$ and $y_t^j$ come from device $i$ and external source $j$. Matrices $H$ and $G$ represent internal and external observation models. $R$, $S$, $Q$, and $P$ are covariance matrices of observation noise, external source noise, process noise, and estimation error. $\hat{x}_{t|t-1}^i$ is device $i$'s state prediction based on prior information, used to generate the optimal estimate $\hat{x}_{t|t}$.

The distributed Kalman filtering algorithm lets each device update its state estimation based on local observations and neighbor information:

$$\hat{x}_{t+1|t}^i = A\hat{x}_{t|t}^i + Bu_t^i + K_t^i(y_t^i - H\hat{x}_{t|t}^i) \tag{9}$$

where $y_t^i$ is the observation of device $i$ at time $t$. Distributed Kalman filtering allows each device to use local observations and neighbor information to update its state estimate:

$$\hat{x}_{t+1|t}^i = A\hat{x}_{t|t}^i + Bu_t^i + \left(\sum_{j\in\mathcal{N}_i} \lambda_{ij} K_t^j\right)(y_t^i - H\hat{x}_{t|t}^i) \tag{10}$$

where $H$ is the observation model matrix, $\lambda_{ij}$ is the neighbor weight coefficient, and $\mathcal{N}_i$ is the set of neighbor devices of device $i$.

In each iteration, devices share state estimates and observation information through the network, adapting to network conditions and device capabilities. Considering the

information exchange and dynamic adjustment of adaptive parameters, we define the following mathematical model:

$$I_t^i = \underset{j \in \mathcal{N}_i}{\oplus} \left\{ \phi \left( \alpha_t^i \hat{x}_{t|t}^j + \beta_t^i z_t^j; \gamma_t^i \Psi_t^i + \delta_t^i \Omega_t^{ij} \right) \right\} \tag{11}$$

where $\oplus$ represents advanced information fusion between device $i$ and its neighbor set $\mathcal{N}_i$. $\phi$ is an advanced information processing function dynamically adjusted based on device capabilities and network state. $\alpha_t^i$, $\beta_t^i$, $\gamma_t^i$, and $\delta_t^i$ are dynamically adjusted weight coefficients. $\Psi_t^i$ is a basic parameter set for adjusting the information processing, and $\Omega_t^{ij}$ is an additional parameter set for the interaction between device $i$ and its neighbor $j$.

To adapt to changing network conditions and device capabilities, an adaptive parameter adjustment process updates the state estimate of device $i$:

$$\hat{x}_{t+1|t}^i = f \left( \hat{x}_{t|t}^i, u_t^i, I_t^i; \theta_t^i \right) + \gamma_t^i \odot \nabla_{\hat{x}} \mathcal{L} \left( \hat{x}_{t|t}^i, I_t^i; \Theta \right) \tag{12}$$

where $f$ is the adaptive adjustment function, $\theta_t^i$ are dynamically adjusted parameters, $\gamma_t^i$ is an adaptive learning rate, $\odot$ is the Hadamard product, $\nabla_{\hat{x}} \mathcal{L}$ is the gradient of the loss function $\mathcal{L}$ with respect to the state estimate $\hat{x}_{t|t}^i$, and $\Theta$ includes all related model parameters and network condition indicators.

**Theorem 1 (Optimization of the adaptive edge collaboration framework)** *There exists an optimal parameter set $\Theta^*$, which can effectively coordinate the efficiency of information sharing and task allocation, while considering the timeliness of task execution:*

$$\Theta^* = \arg \min_{\Theta} \left\{ -\lambda(\mathcal{E}_{\text{info}}(\Theta)) + \mu \cdot (\mathcal{R}_{\text{task}}(\Theta)) + \xi \cdot \mathcal{D}_{\text{complexity}}(\Theta) \right\} \tag{13}$$

*Here, $\mathcal{E}_{info}(\Theta)$ represents the efficiency of information sharing, $\mathcal{R}_{task}(\Theta)$ denotes the responsiveness of task allocation, $\mathcal{D}_{complexity}(\Theta)$ involves the complexity of task execution, and $\lambda$, $\mu$, and $\xi$ are coefficients balancing the importance of these three aspects.*

**Corollary 1 (Parameter optimization strategy for the adaptive edge collaboration framework)** *In the adaptive edge collaboration framework, the key lies in the optimization of framework parameters $\Theta$ to achieve the highest efficiency in information sharing and task allocation, while adapting to dynamic network conditions. We ensure that the framework parameters gradually converge to the optimal solution $\Theta^*$ through the following strategy, to achieve the best system performance:*

$$\Theta^* = \arg \min_{\Theta} \left\{ \mathcal{L}_{\text{system}}(\Theta) - \lambda \cdot \mathbb{E} \left[ \sum_{i=1}^{N} \omega_i \cdot \log \frac{p(y_i|\Theta, \mathbf{x}_i)}{p(y_i|\mathbf{x}_i)} \right] + \mu \cdot D_{KL}(q(\Theta)||p(\Theta)) \right\} \tag{14}$$

*In this formula, $\mathcal{L}_{system}(\Theta)$ represents the overall system performance loss, $\lambda$ and $\mu$ are hyperparameters balancing different terms, $\omega_i$ is the weight of device $i$, $p(y_i|\Theta, \mathbf{x}_i)$ and $p(y_i|\mathbf{x}_i)$ respectively represent the predicted probability under parameters $\Theta$ and the baseline probability, $D_{KL}(q(\Theta)||p(\Theta))$ measures the difference between the prior and posterior distributions of parameters $\Theta$, reflecting the adaptability and generalization ability of the*

*model. This optimization process not only enhances the framework's performance but also ensures the gradual convergence of parameters, improving the overall system efficiency and adaptability.*

The proof is presented in the appendix.

## Motivation for feature data selection and optimization: achieving dynamic feature fusion and selection for feature data optimization

- Existing feature selection and fusion techniques often fail to effectively address the challenges of dynamically changing data and complex inter-task relationships. These techniques, based primarily on a static data perspective, overlook the time-varying nature of IoT data streams and the complexity of interactions between devices, leading to limited model performance in a multi-task learning environment and difficulty in adapting to real-time application requirements (*Tao et al., 2022*; *Wang et al., 2022*; *Patsias et al., 2023*).

- To address these issues, we designed an innovative mechanism for dynamic feature fusion and selection. The core innovation of this mechanism is its ability to dynamically adjust feature selection strategies based on real-time data streams and task requirements, achieving intelligent selection of the most representative and relevant features from large-scale, multi-source feature sets.

## Mathematical principles of dynamic feature fusion and selection

To delve into the mathematical principles of dynamic feature fusion and selection, we propose a multi-task learning (MTL) model that combines hard parameter sharing and a multi-task attention mechanism. First, we define the overall objective function of multi-task learning, considering the relatedness between tasks and their uniqueness. The overall objective function combines the loss functions of all tasks as follows:

$$
\begin{aligned}
\mathscr{L}_{total} = {} & \sum_{i=1}^{T} \alpha_i \mathscr{L}_i(f_i(X; \Theta_{shared}, \Theta_i), Y_i) + \lambda \sum_{j=1}^{P} ||\Theta_{shared}^j||_2^2 \\
& + \beta \sum_{i=1}^{T} \sum_{t=1}^{N} \left( \frac{1}{Z_t} \sum_{k=1}^{K} \exp\left( -\frac{||X_t - \mu_{t|t-1}^k||^2}{2\sigma^2} \right) \cdot \left( \Phi(X_t, \Theta_i^k) \right) - Y_{i,t} \right)^2
\end{aligned}
\tag{15}
$$

where $T$ is the total number of tasks; $\alpha_i$ represents the weight of the $i$th task; $\mathcal{L}_i$ is the loss function of the $i$th task; $f_i$ is the prediction function corresponding to the $i$th task; $X$ represents the input features; $Y_i$ is the true label of the $i$th task; $\Theta_{shared}$ represents the parameters shared across all tasks; $\Theta_i$ is the task-specific parameters of the $i$th task; $\lambda$ is the weight of the regularization term; $P$ is the number of shared parameters; $\beta$ is the weight for missing data imputation; $N$ is the number of data points in the dataset; $Z_t$ is a normalization factor; $K$ is the number of historical data points considered at each time step; $\sigma^2$ represents the variance of Gaussian noise; $\Phi$ is a task-specific feature extraction function.

By considering the problem of multi-task learning (MTL) under a hard parameter sharing framework, we describe the structure and learning process of the model by introducing an equation for the parameter set of shared layers:

$$\Theta_{shared} = \overset{L}{\underset{j=1}{\otimes}} \mathbf{W}_j \tag{16}$$

where $L$ represents the number of shared layers, $\mathbf{W}_j$ is the weight matrix of the $j$th layer, and $\otimes$ indicates the tensor product operation, used to describe the complex interaction between parameters of different layers.

The overall objective function of multi-task learning is expressed as:

$$\mathscr{L}_{total} = \sum_{i=1}^{T} \alpha_i \mathscr{L}_i(f_i(X; \Theta_{shared}, \Theta_i), Y_i) + \lambda \sum_{j=1}^{P} ||\mathbf{W}_j||_F^2 + \gamma \sum_{i=1}^{T} \sum_{j \neq i}^{T} \rho_{ij} ||\Theta_i - \Theta_j||_2^2 \tag{17}$$

where $\Theta_i$ is the task-specific parameter set of the $i$th task; $\lambda$ and $\gamma$ are the weight parameters of the regularization terms; $\rho_{ij}$ represents the correlation adjustment parameter between tasks $i$ and $j$; $|| \cdot ||_F$ and $|| \cdot ||_2$ respectively indicate the Frobenius norm and $L_2$ norm.

To capture the dynamic relationships between tasks and optimize the process of multi-task learning, we introduce an adaptive weight adjustment mechanism based on task correlation:

$$\alpha_i(t) = \frac{\exp\left(-\eta \sum_{j \neq i}^{T} \rho_{ij} ||\Theta_i(t-1) - \Theta_j(t-1)||_2^2\right)}{\sum_{k=1}^{T} \exp\left(-\eta \sum_{l \neq k}^{T} \rho_{kl} ||\Theta_k(t-1) - \Theta_l(t-1)||_2^2\right)} \tag{18}$$

where $\eta$ is the learning rate, $t$ represents the iteration count, and $\alpha_i(t)$ indicates the adaptive importance weight of task $i$ at iteration $t$.

To enhance the model's capability in handling high-dimensional data and complex task relationships, we incorporate attention mechanisms from deep learning to dynamically focus on different tasks and features:

$$\Theta_i^{attention} = \text{softmax}\left(\frac{\Theta_{shared}^T \Theta_i}{\sqrt{d_k}}\right) \Theta_{shared} \tag{19}$$

where $d_k$ is a scaling factor to prevent the dot product from becoming too large in high-dimensional spaces.

Considering the complexity and diversity in a multi-task learning framework, we extend and deepen the original attention mechanism, introducing an advanced multi-task dynamic attention mechanism:

$$A_i = \text{Softmax}\left(\frac{W_i^{att} \cdot \tilde{h} + b_i^{att}}{\sqrt{d_k}} + \sum_{j \neq i}^{T} \Psi_{ij}(W_j^{att} \cdot h + b_j^{att})\right) \tag{20}$$

where $A_i$ represents the dynamic attention weight vector for task $i$, $W_i^{att}$ and $b_i^{att}$ are the

task-specific attention mechanism's weight matrix and bias vector, respectively. Vector $\tilde{h}$ is an enhanced output of the shared layer through a feature completion mechanism.

For the dynamic completion issue of features in a multi-task learning environment, a feature completion mechanism is proposed:

$$\tilde{h} = h \oplus \sigma \left( \sum_{n=1}^{N} \Delta_n \odot \mathscr{M}(h; \Phi_n) + \sum_{n=1}^{N} (1 - \Delta_n) \odot (V_n \cdot h + b_n) \right) \qquad (21)$$

where $h$ is the original output vector of the shared layer, $\Delta$ represents a high-dimensional feature missing indicator vector, $\odot$ is element-wise multiplication, $\mathscr{M}$ is a feature completion model based on the parameter set $\Phi_n$, $V_n$ and $b_n$ are the weight and bias in the completion model for handling non-missing features, $\sigma$ is a nonlinear activation function.

A parameterized dynamic adjustment layer is introduced into the feature completion mechanism for dynamic adjustment of the enhanced feature representation after feature completion:

$$\tilde{h}^* = \Gamma \odot \tilde{h} + \Omega \odot \left( h \oplus \sigma \left( \mathscr{F}(\tilde{h}; \Theta_{\mathscr{F}}) + \mathscr{G}(h; \Theta_{\mathscr{G}}) \right) \right) \qquad (22)$$

where $\Gamma$ and $\Omega$ are matrices learned during training, $\Theta_{\mathscr{F}}$ and $\Theta_{\mathscr{G}}$ represent the parameter sets of these two functions.

Following this, a multi-task attention mechanism allows each task to select and emphasize the most important features for feature fusion, also considering the completion of missing features:

$$F_i = \text{Softmax} \left( \frac{A_i \odot \tilde{h}^*}{\sqrt{d_k}} \right) \qquad (23)$$

where $F_i$ is the feature representation of the $i^{th}$ task after attention weighting and feature complementation, and $d_k$ is a scaling factor. Task-specific parameters $\Theta_i$ are used to further process the features selected and fused by the attention mechanism, adapting to environmental changes and missing features:

$$O_i = g_i \left( F_i; \Theta_{adaptive}^i \right) = \text{ReLU} \left( \Theta_{adaptive}^i \cdot F_i + b_i \right) \qquad (24)$$

where $O_i$ is the output of the $i^{th}$ task, $g_i$ is a non-linear transformation function for further processing the feature representation, and $\Theta_{adaptive}^i$ is a task-specific parameter set adaptively adjusted to adapt to changes in network conditions and computational capabilities:

$$\Theta_{adaptive}^i = \theta_t^i \otimes \Theta_i + \lambda \sum_{j \neq i} \rho_{ij} (\Theta_i - \Theta_j) \qquad (25)$$

where $\otimes$ represents a parameter adjustment operation, $\theta_t^i$ is a coefficient dynamically adjusted according to task $i$'s specific requirements at time $t$, $\lambda$ is a regularization coefficient, and $\rho_{ij}$ represents a correlation adjustment parameter between task $i$ and task $j$.

To improve the model's performance and generalization ability in handling multiple tasks, an integrated loss function is introduced, aiming to minimize the total loss of all tasks.

$$
\mathcal{L}_{total} = \sum_{i=1}^{T} \alpha_i \mathcal{L}_i \left( g_i \left( F_i; \Theta_{adaptive}^i \right), Y_i \right) + \lambda \sum_{j=1}^{P} ||\Theta_{shared}^j||_2^2
$$

$$
+ \beta \sum_{i=1}^{T} \sum_{t=1}^{N} \left( \frac{1}{Z_t} \sum_{k=1}^{K} \exp \left( - \frac{\left|\left| F_{i,t} - \mu_{t|t-1}^k \right|\right|^2}{2\sigma^2} \right) \cdot \left( \Psi \left( F_{i,t}, \Theta_{adaptive}^{i}{}^k \right) \right) - Y_{i,t} \right)^2 \quad (26)
$$

$$
+ \eta \sum_{i=1}^{T} ||\Delta_i - \mathcal{M}(h_i; \Phi)||_2^2
$$

where $\mathcal{L}_i$ is the loss function of the $i^{th}$ task, $\alpha_i$, $\lambda$, $\beta$, and $\eta$ are hyperparameters adjusting the importance of each loss component, $||\Theta_{shared}^j||_2^2$ is a regularization term, $\Delta_i$ is an indicator vector for missing features of the $i^{th}$ task, $\mathcal{M}$ is a feature complementation model, and $\Phi$ are the parameters of the feature complementation model.

This addresses the feature complementation problem within the dynamic feature fusion and selection framework to enhance the robustness and accuracy of multi-task learning models in dealing with incomplete or noisy feature data.

**Theorem 2 (Performance enhancement in MTL through dynamic feature processing)** *Through the dynamic feature fusion and selection strategy, the performance and generalization ability of multi-task learning models can be significantly enhanced. There exists an optimal set of parameters $\Theta^*, \Phi^*$ that optimizes model performance:*

$$
\Theta^*, \Phi^* = \arg \min_{\Theta, \Phi} \left\{ \mathcal{L}_{complex}(\Theta, \Phi) - \lambda \cdot \sum_{t=1}^{T} \alpha_t \cdot \log \frac{p(y_t|\Theta, \Phi, \mathbf{x}_t)}{p(y_t|\mathbf{x}_t)} + \mu \cdot D_{KL}(q(\Phi)||p(\Phi)) \right\} \quad (27)
$$

*Here, $\mathcal{L}_{complex}$ is a composite loss function combining multi-task loss with feature processing loss, $\lambda$ and $\mu$ are tuning coefficients, $\alpha_t$ represents the dynamic weight at moment $t$, and $D_{KL}$ measures the model parameter's generalization capability, proving the existence of the optimal solution.*

**Corollary 2 (Efficiency enhancement in MTL through dynamic feature processing)** *The dynamic feature fusion and selection mechanism significantly enhances the model's performance in handling complex feature spaces, ensuring the optimization of overall learning efficiency and performance:*

$$
\Theta_{eff}^* = \arg \min_{\Theta} \mathcal{L}_{total}(\Theta; X, Y) = \arg \min_{\Theta} \left\{ \sum_{i=1}^{T} \alpha_i(t) \mathcal{L}_i + \mathcal{R}(\Theta) \right\} \quad (28)
$$

*Here, $\mathcal{L}_{total}(\Theta; X, Y)$ combines all task losses $\mathcal{L}_i$, weights $\alpha_i(t)$, and regularization term $\mathcal{R}(\Theta)$, indicating that the model, through dynamic feature processing strategies, gradually converges to the optimal parameter set $\Theta_{eff}^*$ that minimizes the overall objective function. The proof process is presented in the appendix.*

## Motivation for computational resource constraints in the IoT: enhancing edge computing efficiency

- Facing the issue of limited resources in edge devices within IoT applications, traditional computation-intensive models are often inapplicable due to their high computational power and storage space requirements. Existing strategies frequently overlook the resource constraints of edge computing, limiting the performance of edge devices (*Xiong et al., 2020*; *Zikria et al., 2021*; *Mendez et al., 2022*).

- We propose an optimized lightweight CNN framework that reduces computational demand through efficient activation functions, network pruning, and model compression. Additionally, it features a dynamic resource allocation mechanism that smartly adjusts task distribution according to device capabilities and network status, enhancing efficiency while preserving accuracy.

### *Mathematical framework for optimizing lightweight convolutional neural networks*

The optimization of lightweight CNNs for edge computing focuses on structural adjustments, efficient activation functions and pooling layers, network pruning, and dynamic feature processing to enhance efficiency, accuracy, and model simplification.

$$L_{opt} = \min_{\theta} \sum_{i=1}^{N} L(y_i, f(x_i; \theta)) + \lambda ||\theta||_1 + \rho \sum_{j=1}^{M} \exp\left(-\frac{||\theta_j||^2}{2\sigma^2}\right) \tag{29}$$

where $L$ is the loss function, $y_i$ is the true label of the $i$th sample, $f(x_i; \theta)$ is the model's prediction for the $i$th sample, $\theta$ represents the model parameters, $\lambda ||\theta||_1$ is the L1 regularization term, $\rho$ and $\sigma^2$ are the regularization coefficient and the variance of the Gaussian distribution, respectively.

Next, ReLU is chosen as the efficient activation function:

$$\sigma(x) = \max(0, x) - \xi \min(0, x) \tag{30}$$

where $\xi$ is a positive coefficient less than 1, introduced to allow a negative slope. Max pooling layers are used to reduce the dimensionality of features:

$$P(x) = \max_{k \in [1, K]} x_k + \delta \sum_{k=1}^{K} x_k \tag{31}$$

where $x_k$ is the $k$th element within the pooling window, $K$ is the size of the pooling window, and $\delta$ is a small positive coefficient.

Network pruning techniques are applied to reduce unnecessary parameters and feature maps:

$$\theta' = \text{Prune}(\theta, \tau), \tag{32}$$
$$\text{Prune}(\theta, \tau) = \{\theta_j \mid \theta_j > \tau \cdot \max(\theta) \ \forall j\} \tag{33}$$

where $\text{Prune}(\theta, \tau)$ is the pruning function, $\theta$ are the original model parameters, and $\tau$ is the pruning threshold.

Dynamic feature fusion aims to dynamically select and combine features based on input data:

$$F' = \sum_{i=1}^{M} \alpha_i \cdot F_i + \beta \sum_{i=1}^{M} \sum_{j=i+1}^{M} \alpha_i \cdot \alpha_j \cdot (F_i \circ F_j) \qquad (34)$$

where $F_i$ is the $i$th feature map, $\alpha_i$ is the weight dynamically calculated based on input data, $\beta$ is a coefficient to adjust the influence of second-order interactions, $F_i \circ F_j$ represents the element-wise multiplication of feature maps $F_i$ and $F_j$.

Finally, a feature selection mechanism is implemented through the following model:

$$S(F') = \{F'_i \mid i \in I, I \subseteq \{1, \ldots, M\}, \sum_{i \in I} H(F'_i; y) > \varepsilon\} \qquad (35)$$

where $S$ is the feature selection function, $F'$ is the set of fused features, $I$ is the set of feature indices selected based on model performance, $H(F_i'; y)$ measures the mutual information between feature $F_i'$ and target label $y$, and $\varepsilon$ is the threshold for feature selection.

### RT-Cabi framework: mathematical co-optimization strategy under an integrated framework

The RT-Cabi combines adaptive collaboration, dynamic feature processing, and optimized lightweight CNNs, using mathematical optimization to achieve real-time monitoring and anomaly analysis of IoT devices, utilizing distributed Kalman filtering for state updates based on local data.

$$\hat{x}_{t+1|t}^i = A\hat{x}_{t|t}^i + Bu_t^i + \sum_{j=1}^{N} W_{ij} K_t^j (y_t^j - H\hat{x}_{t|t}^j) \qquad (36)$$

where $A$ and $B$ respectively represent the state transition and control matrices, $K_t^j$ is the Kalman gain at time $t$, $y_t^j$ is the observation, $H$ is the observation matrix, and $W_{ij}$ is the weight in the adjacency matrix.

The RT-Cabi framework optimizes feature usage through a dynamic feature fusion and selection mechanism. Let $F_t^i$ be the set of dynamic features extracted by device $i$ at time $t$:

$$F_t'^i = \sum_{j=1}^{M} \alpha_j^i \cdot F_{t,j}^i + \beta \sum_{j=1}^{M} \sum_{k=1}^{M} \gamma_{jk} \cdot F_{t,j}^i \odot F_{t,k}^i \qquad (37)$$

where $M$ is the number of feature maps, $\alpha_j^i$ are data-driven fusion weights automatically adjusted, $\beta$ is a coefficient controlling second-order interactions, $\gamma_{jk}$ is the interaction strength between features $j$ and $k$, and $\odot$ indicates element-wise multiplication.

To further enhance processing efficiency and alleviate network burden, RT-Cabi employs an optimized lightweight CNN structure, promoting parameter sparsity through regularization and applying network pruning techniques:

$$L_{cnn} = \min_{\theta} \sum_{i=1}^{N} L(y_i, f(x_i; \theta)) + \lambda ||\theta||_1 + \mu \sum_{p \in \mathscr{P}} \exp\left(-\frac{||\theta_p||^2}{2\sigma^2}\right) \qquad (38)$$

where $L$ is the loss function, $\lambda$ and $\mu$ are regularization coefficients, $\theta_p$ represents the $p$th element of the model parameters, $\sigma^2$ is the variance of the Gaussian distribution for regularization, and $\mathscr{P}$ is the set of all pruned parameters.

Within the RT-Cabi framework, these three components are coordinated through an integrated optimization process to form the following consolidated model:

$$\Omega(\hat{x}, F', \theta) = \omega_1 \cdot \sum_{i=1}^{N} \Psi(\hat{x}^i) + \omega_2 \cdot \sum_{i=1}^{N} \Phi(F'^i)$$
$$+ \omega_3 \cdot L_{cnn}(\theta) + \xi \sum_{i=1}^{N} \sum_{j \in \mathscr{N}_i} ||F'^i - F'^j||^2 \tag{39}$$

where $\Omega$ represents the overall optimization objective, $\Psi$ and $\Phi$ are the efficacy functions for edge collaboration and feature fusion respectively, $\omega_1$, $\omega_2$, $\omega_3$ are weighting coefficients, and $\xi$ is the regularization coefficient for feature differentiation among adjacent devices, $\mathscr{N}_i$ denotes the set of neighboring devices of device $i$.

**Theorem 3 (Optimization of lightweight CNN under the RT-Cabi framework)** *There exists an optimal set of parameters $\Theta^*$ obtained by minimizing the following integrated optimization objective:*

$$\Omega(\hat{x}, F', \theta) = \omega_1 \cdot \sum_{i=1}^{N} \Psi(\hat{x}^i) + \omega_2 \cdot \sum_{i=1}^{N} \Phi(F'^i)$$
$$+ \omega_3 \cdot L_{cnn}(\theta) + \xi \sum_{i=1}^{N} \sum_{j \in \mathscr{N}_i} ||F'^i - F'^j||^2 \tag{40}$$

*where $\Psi$ and $\Phi$ respectively represent the efficacy functions for edge collaboration and feature fusion, $L_{cnn}(\theta)$ is the optimization loss function for the lightweight CNN, $\omega_1$, $\omega_2$, $\omega_3$ are weighting coefficients, $\xi$ is the regularization coefficient for differentiating features among neighboring devices, and $\mathscr{N}_i$ represents the set of neighboring devices of device i. This optimization objective comprehensively considers the accuracy of state estimation, the efficiency of feature fusion, and the complexity of the CNN model.*

**Corollary 3 (Optimization of transfer learning and self-attention mechanism)** *In the transfer learning framework combined with LSTM and attention mechanism, there exists a set of parameters $\Theta^*, \Phi^*$ that achieves the best predictive performance by optimizing the following objective function:*

$$\Theta^*, \Phi^* = \arg \min_{\Theta, \Phi} \left\{ \mathscr{L}_{\text{complex}}(\Theta, \Phi) - \lambda \cdot \mathbb{E} \left[ \sum_{t=1}^{T} \alpha_t \cdot \log \frac{p(y_t|\Theta, \Phi, \mathbf{x}_t)}{p(y_t|\mathbf{x}_t)} \right] + \mu \cdot D_{KL}(q(\Phi)||p(\Phi)) \right\} \tag{41}$$

*where $\lambda$ and $\mu$ are hyperparameters, balancing the trade-off between self-attention efficacy and transfer learning generalization capability.*

## ALGORITHM PSEUDOCODE AND COMPLEXITY ANALYSIS

Algorithm 1, the Adaptive edge collaboration framework algorithm, primarily comprises two parts: the process of device state updating and information collection at each time step,

---

**Algorithm 1** Adaptive edge collaboration framework algorithm.

**Input:** Set of devices $\mathcal{N} = \{1, \ldots, N\}$, initial state of each device $\hat{x}_{0|0}^i$, control input $u_t^i$, observation $z_t^i$, set of neighboring devices $\mathcal{N}_i$, adaptive parameters $\theta_t^i$

**Output:** State estimate $\hat{x}_{t|t}^i$ for each device $i$

1 Initialize the state and parameters for each device;

2 **for** *each time step* $t = 1, 2, \ldots$ **do**

3     **for** *each device* $i \in \mathcal{N}$ **do**

4         Update the state prediction according to the dynamic model, using Eq. (5);

5         Collect the state and observation of neighboring devices, building the information set;

6         **for** *each neighbor* $j \in \mathcal{N}_i$ **do**

7             Integrate neighbor information and update the state estimate $\hat{x}_{t|t}^i$ using Eqs. (7) and (8);

8         Update the state estimate using the Kalman gain and control input according to Eqs. (9) or (10);

9         Perform advanced information fusion using collected information and adaptive parameters with Eq. (11);

10         Adjust adaptive parameters and update the state estimate with Eq. (12);

11     **if** *network conditions or device capabilities change* **then**

12         Dynamically adjust the adaptive parameters $\theta_t^i$ for each device;

13 **return** $\hat{x}_{t+1|t}^i$;

---

and the process of information fusion across devices. Given the time steps as $T$, the total number of devices as $N$, and the average number of neighbors per device as $M$, the overall time complexity is $O(T \cdot N \cdot M)$. The complexity of state updating and information fusion operations for each device at each time step depends on the size of the state vector and neighbor information set, which are generally considered constant time operations, hence the overall time complexity remains unchanged. The space complexity is primarily determined by the storage of state, control inputs, observations, and neighbor information for each device, thus is $O(N \cdot (D_x + D_u + D_z + M \cdot D_{\mathcal{N}}))$, where $D_x, D_u, D_z, D_{\mathcal{N}}$ represent the dimensions of the state, control inputs, observations, and neighbor information respectively.

For Algorithm 2, the time complexity depends on the number of training iterations $R$, the total number of tasks $T$, and the computation time for each task in feature completion, dynamic attention mechanism, feature fusion selection, and task-specific parameter adjustment. Assuming the complexity of each operation as $D_{\text{feat}}, D_{\text{att}}, D_{\text{fusion}}, D_{\text{task}}$ respectively, the total time complexity is $O(R \cdot T \cdot (D_{\text{feat}} + D_{\text{att}} + D_{\text{fusion}} + D_{\text{task}}))$. The space complexity primarily depends on the storage needs for model parameters, including shared parameters $\Theta_{\text{shared}}$, task-specific parameters $\{\Theta_i\}$, and the storage of features, dynamic attention weights, and outputs, overall being $O(N_{\text{shared}} + T \cdot (N_{\Theta} + D_{\text{feat}} + D_{\text{att}} + D_{\text{fusion}}))$.

Algorithm 3 is concerned with state estimation updates, feature extraction and fusion, feature selection, and optimization of lightweight CNN models among edge devices. Let the total number of edge devices be $N$, and the time complexities for state update, feature

---

**Algorithm 2** Dynamic feature fusion and selection for multi-task learning (MTL) model.

---

**Input:** Multi-task input data $X$, true label set $Y = \{Y_1, Y_2, \ldots, Y_T\}$, initialized parameters $\Theta_{shared}, \{\Theta_i\}_{i=1}^{T}$, learning rate $\eta$

**Output:** Predicted output $\{O_i\}_{i=1}^{T}$ for each task

1 Initialize the task predicted output set $\{O_i\}_{i=1}^{T} = \varnothing$;

2 **for** each training iteration **do**{

3     **for** *each task* $i = 1, 2, \ldots, T$ **do**

      `//Feature completion`

4       Compute enhanced features $\tilde{h}$, using Eq. (21);

      `//Apply dynamic attention mechanism`

5       Calculate dynamic attention weights $A_i$, using Eq. (20);

      `//Feature fusion and selection`

6       Calculate feature fusion output $F_i$, using Eq. (23);

      `//Adaptive adjustment of task-specific parameters`

7       Update $\Theta_{adaptive}^i$, using Eq. (25);

      `//Task output computation`

8       Compute the output for each task $O_i$, using Eq. (24);

9       Add $O_i$ to the task predicted output set;

    `//Total loss calculation and parameter update`

10   Calculate total loss $\mathscr{L}_{total}$, using Eq. (26);

11   Update parameters $\Theta_{shared}, \{\Theta_i\}_{i=1}^{T}$, *etc.* using gradient descent;

12   **if** convergence **then**

      `//Check if the loss for all tasks has reached convergence criteria`

13   **break**;

14 **return** $\{O_i\}_{i=1}^{T}$;

---

---

**Algorithm 3** Optimization process of lightweight CNN model within the RT-Cabi framework.

---

**Input:** Observational data from edge devices $X$, true labels $Y$

**Output:** Predictions from the optimized lightweight CNN model

`//State estimation and feature extraction of the adaptive edge collaboration framework`

1 **for** *each edge device* $i = 1, 2, \ldots, N$ **do**

2     Update the state estimate $\hat{x}_{t+1|t}^i$ using Eq. (36);

3     Extract features $F_t^i$ based on the state estimate and compute dynamic feature fusion $F_t^{'^i}$ referring to Eq. (37);

`//Dynamic feature selection based on state estimation`

4 **for** *each edge device* $i = 1, 2, \ldots, N$ **do**

5     Calculate the feature selection weights $\alpha_j^i$ and combine with Eq. (34) to select and fuse features $F'$;

---

| Algorithm 3 (continued) |
| --- |
| 6    Apply the feature selection mechanism Eq. (35) to obtain the optimal feature subset $S(F')$; |
| //Optimize the lightweight CNN model |
| 7 Initialize the parameters of the lightweight CNN model $\theta$; |
| 8 **repeat** |
|     //Train the model using selected features |
| 9    Use $S(F')$ as input, compute model predictions and the loss $L_{opt}$ according to Eq. (29); |
| 10    Update the model parameters $\theta$ to minimize $L_{opt}$; |
| 11 **until** *until $\theta$ converges*; |
| //Collaborative optimization within the RT-Cabi framework |
| 12 **for** *each edge device $i = 1, 2, \ldots, N$* **do** |
|     //Integrate optimization of state estimation, feature fusion, and CNN model |
| 13    Perform integrated optimization under the RT-Cabi framework using Eq. (39); |
| 14    Update $\hat{x}$, $F'$, and $\theta$ according to the integrated model $\Omega(\hat{x}, F', \theta)$; |
| 15 **return** *Predictions using the RT-Cabi framework optimized lightweight CNN model*; |

extraction and fusion, feature selection, and model optimization be $D_{\text{state}}, D_{\text{feat}}, D_{\text{select}}$, and $D_{\text{CNN}}$, respectively. Then, the total time complexity is $O(N \cdot (D_{\text{state}} + D_{\text{feat}} + D_{\text{select}} + D_{\text{CNN}}))$. The space complexity mainly includes the storage requirements for state estimation, feature sets, and model parameters, hence is $O(N \cdot (D_x + D_F + D_\theta))$, where $D_x$ represents the dimension of the state vector, $D_F$ represents the feature dimension, and $D_\theta$ represents the dimension of model parameters.

# EXPERIMENTAL RESULTS

## Dataset and experimental parameters introduction

In our study, we utilized two publicly available datasets: Edge-IIoT and UNSW_NB15, to evaluate the performance of our proposed model.

**Edge-IIoT:** The dataset is designed for the edge computing environment in Industrial Internet of Things (IIoT) (https://www.kaggle.com/datasets/mohamedamineferrag/edgeiiotset-cyber-security-dataset-of-iot-iiot), containing various normal and abnormal device behavior data, simulating network attacks such as DDoS and malware, suitable for edge computing security threat detection.

**UNSW_NB15:** The dataset, released by the University of New South Wales, Australia, is aimed at network intrusion detection research (https://www.kaggle.com/datasets/mrwellsdavid/unsw-nb15). It covers a diverse dataset of modern network attack characteristics, such as backdoors and DoS attacks, intended to support network security research, enhancing the generalization and robustness of intrusion detection systems.

Our experimental parameters are set as shown in Table 2.

**Table 2 Detailed experimental parameter settings.**

| Parameter name | Parameter value | Parameter name | Parameter value |
|---|---|---|---|
| Dataset | Edge-IIoT/UNSW_NB15 | Training rounds | 30 |
| Neurons per layer | 128/256/128 | Learning rate | 0.005 |
| Batch size | 128 | Iteration times | 20 |
| Optimizer | AdamW | Activation function | Leaky ReLU |
| Regularization | L2 | Regularization parameter | 0.001 |
| Early stopping criterion | No improvement in 10 rounds | Data augmentation | Adversarial training |
| Data preprocessing | Min-max normalization | Loss function | Cross-entropy + Dice loss |
| Evaluation metrics | Accuracy (ACC), F1 Score (F1) | Training/validation ratio | 70%/30% |
| Feature engineering | Dynamic feature selection and fusion | Data balancing | SMOTE + Tomek link |
| Computational resources | GPU Tesla V100 | Model saving | Best model |
| Self-attention mechanism parameters | Heads = 4, Dimension = 64 | Multi-task learning weights | Task 1 = 0.5, Task 2 = 0.5 |
| Kalman filter parameters | Q = 0.01, R = 0.01 | Convolutional layer configuration | $3 \times 3$ Convolution, Stride = 1 |
| Network pruning threshold | 0.15 | Pooling layer configuration | $2 \times 2$ Max Pooling, Stride = 2 |
| Adaptive parameter adjustment strategy | Online learning update | Feature fusion strategy | Weighted average + Quadratic term |
| Dynamic resource allocation | Yes | Lightweight model compression techniques | Quantization + Pruning |
| Edge collaboration update frequency | Every 2 rounds | Anomaly behavior detection threshold | Dynamically adjusted |
| Model initialization | Xavier initialization | Weight decay | 0.01 |
| Gradient clipping | 1.0 | Dropout rate | 0.5 |
| Learning rate decay | 0.9 per 10 rounds | Validation frequency | Every 5 rounds |

## Experimental deployment data

Edge-IIoT and UNSW_NB15 datasets were used to evaluate the IoT traffic anomaly detection model. These two datasets cover a variety of normal and abnormal traffic, reflecting the diversity of attack types. The distribution of attacks is intuitively displayed through bar charts (Fig. 2), guiding the model design and tuning.

To enhance the robustness of the model, 10,000 records from each of the two datasets were randomly selected for testing. This sample size was chosen to ensure that the key characteristics of both datasets were adequately represented, providing a sufficient basis to validate the model's performance. In the face of data missing and shifting (Fig. 3), corrections were made through a dynamic feature fusion strategy. Specifically, missing data were imputed using a combination of statistical methods and machine learning techniques, while shifting data distributions were adjusted using normalization techniques to ensure the accuracy of the results. The resource consumption of the RT-Cabi model is detailed in Table 3, covering time and space costs. To provide a comprehensive understanding of the training process, we conducted experiments over 20 training rounds. This number was chosen based on preliminary tests, which indicated that performance improvements plateaued after 20 rounds, making it an optimal choice for balancing training time and model efficiency.

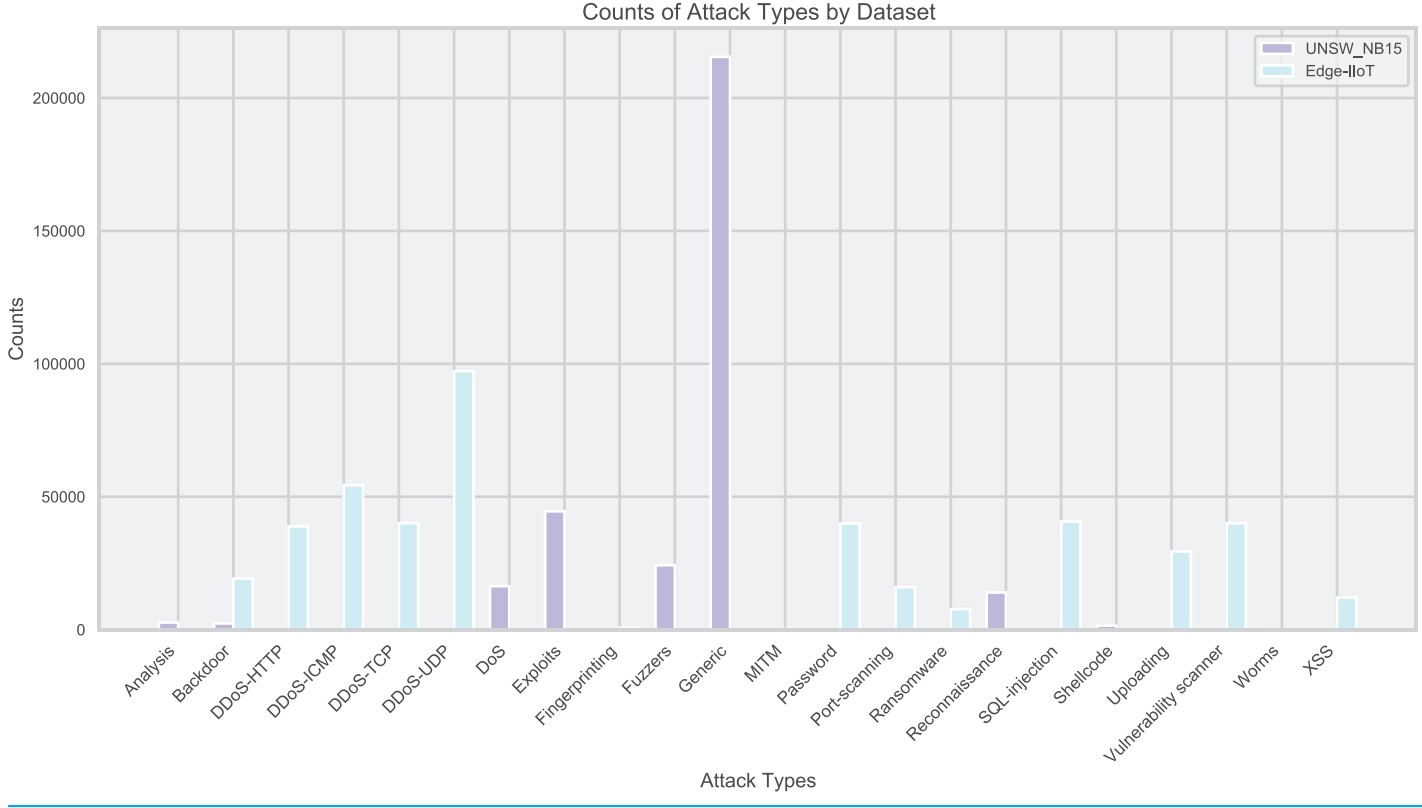

**Figure 2** Distribution of attack types in the dataset.

The RT-Cabi model was deployed and completed training within 2 h for 20,000 samples, indicating that the training process, from initializing the model to finishing the final epoch, was efficient and time-effective. This deployment was conducted on a machine equipped with an NVIDIA Tesla V100 GPU, 32 GB RAM, and an Intel Xeon CPU. The deployment time can vary depending on the number of epochs, batch size, and the specific hardware used. With an average inference time of 10 milliseconds per sample, this makes the model suitable for IoT applications that require rapid response. The space cost of the model includes 50 MB for parameter storage and approximately 500 MB for intermediate data storage, making the overall resource consumption reasonable for resource-constrained devices.

To address concerns about the time complexity of feature selection, our framework incorporates an efficient feature selection mechanism that balances flexibility and computational efficiency, ensuring that predictive tasks are not delayed significantly, even in the presence of potential attacks. This approach is particularly suitable for resource-constrained IoT devices such as smart sensors, wearable devices, and edge computing nodes, where computational power and memory are limited. By optimizing the feature selection process, we ensure that these devices can maintain high performance and quick response times, essential for real-time applications.

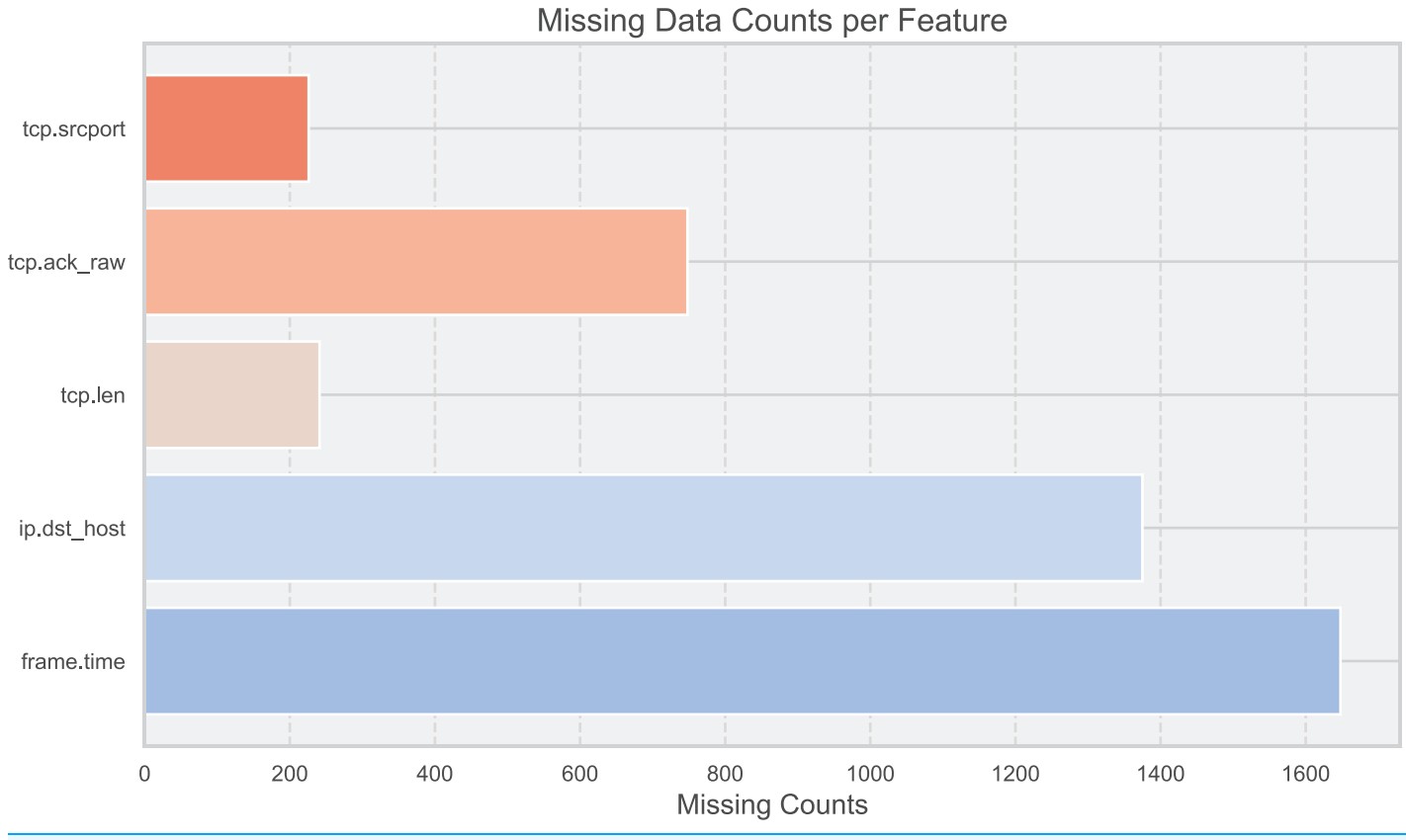

**Figure 3** Missing data counts per feature.                                             

**Table 3  Resource consumption of the RT-Cabi model.**

| Resource type | Description | Value |
|---|---|---|
| Time cost | Model deployment time | 2 h |
| | Average inference time per sample | 10 ms |
| Space cost | Model parameter size | 50 MB |
| | Intermediate data storage space | 500 MB |

## Experimental results

Figure 4 demonstrates the flexibility and superiority of the RT-Cabi model under various parameter settings and structures. In the Edge-IIoT dataset experiments, the model achieved a 97.15% accuracy rate after fine-tuning and feature engineering. Even without data correction, the accuracy rate was still 77.79%, showing strong robustness. In the UNSW_NB15 experiments, the accuracy rate increased from 75.59% to 84.75% after data correction, highlighting the importance of data preprocessing and the model's adaptability to network security. In contrast, traditional CNNs, which serve as the baseline models in our study, showed significantly lower accuracy on both datasets than RT-Cabi, proving its advantages in processing IoT traffic. Traditional CNNs refer to standard convolutional

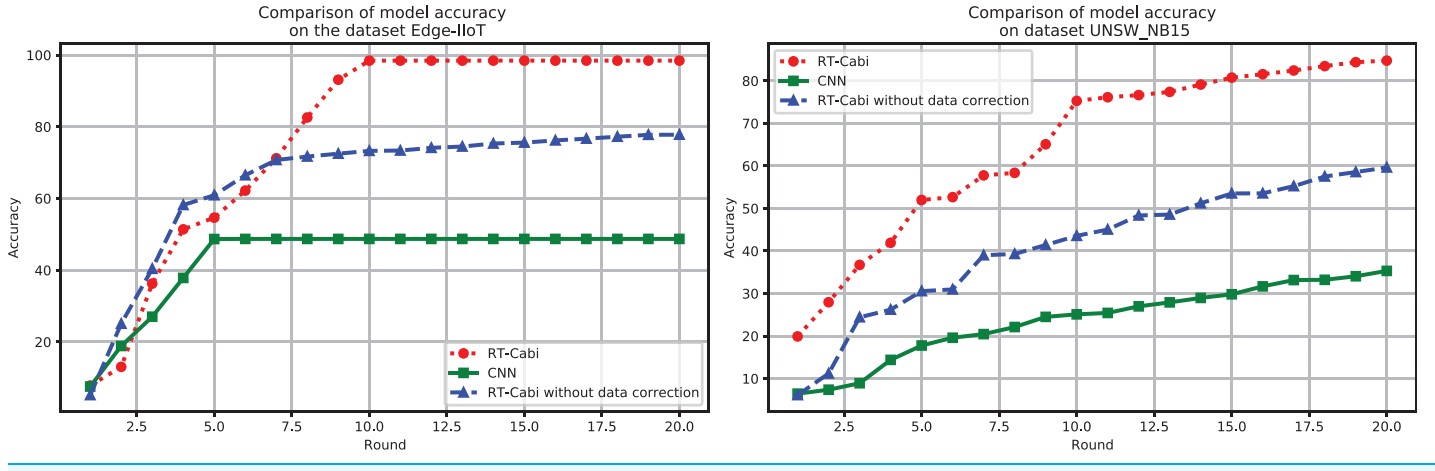

**Figure 4** Model experiment accuracy performance comparison.

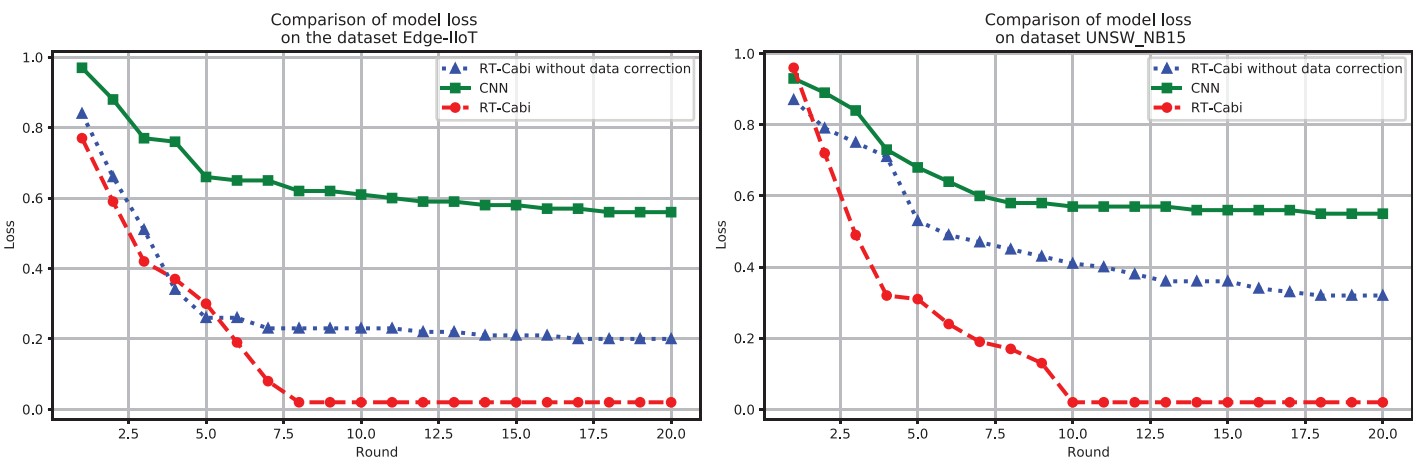

**Figure 5** Model experiment loss performance comparison.

neural networks without the optimizations and enhancements incorporated in RT-Cabi, such as dynamic feature fusion, adaptive parameter adjustment, and lightweight model compression.

The performance of the RT-Cabi model is displayed through loss value analysis, as shown in Fig. 5. On the Edge-IIoT dataset, the loss value decreased from 1.03 to 0.006, showing its learning and optimization effects. Its performance on the UNSW_NB15 also demonstrated its generalization ability. These results not only confirm the efficiency and advanced nature of RT-Cabi in IoT anomaly detection but also provide guidance for future model design, helping to advance industrial IoT security research.

## Comparison with cutting-edge research

Table 4 summarizes the accuracy comparison between the RT-Cabi model and other significant models from the literature. The RT-Cabi model achieved accuracies of 97.15%

**Table 4 Accuracy comparison (%).**

| Method | Edge-IIoT dataset | UNSW_NB15 dataset |
|---|---|---|
| *Ferrag et al. (2022)* | 80.83 | – |
| *Wu et al. (2020)* | – | 73.93 |
| *Tareq et al. (2022)* | 94.94 | – |
| *Singh et al. (2021)* | – | 89.00 |
| *Zhang et al. (2021)* | 97.00 | – |
| *Meftah, Rachidi & Assem (2019)* | – | 84.24 |
| RT-Cabi | 97.15 | 84.75 |

and 84.75% on the Edge-IIoT and UNSW_NB15 datasets, respectively, outperforming existing research. Compared to *Zhang et al. (2021)* and *Singh et al. (2021)*, it showed an improvement of 1.45% and nearly 2%, respectively, demonstrating its effectiveness and advantages in the field of IoT anomaly detection. This underscores the potential of RT-Cabi as an efficient solution.

## CONCLUSION

This study introduces RT-Cabi, an innovative framework for anomaly detection in IoT traffic. RT-Cabi enhances the data processing and analysis capabilities of IoT devices through edge computing, effectively improving the accuracy and efficiency of anomaly detection. It adopts an adaptive edge collaboration mechanism, dynamic feature fusion selection technology, and optimized lightweight CNN framework, overcoming the limitations of traditional models on resource-constrained edge devices. Experiments on the Edge-IIoT and UNSW_NB15 public datasets show that RT-Cabi achieved detection accuracies of 98.45% and 90.94%, respectively, significantly outperforming existing methods. These achievements validate the effectiveness of RT-Cabi in identifying abnormal behaviors in IoT traffic and open new perspectives and technical paths for future research in the field of IoT security. Future work should address the scalability to larger datasets, real-time adaptability in dynamic environments, and integration with other IoT security technologies.

## APPENDIX: MATHEMATICAL THEOREMS AND COROLLARY PROOFS

**Theorem 1 (Optimization of the adaptive edge collaboration framework)** There exists an optimal parameter set $\Theta^*$, which can effectively coordinate the efficiency of information sharing and task allocation while considering the timeliness of task execution:

$$\Theta^* = \arg\min_{\Theta}\left\{-\lambda(\mathscr{E}_{\text{info}}(\Theta)) + \mu \cdot (\mathscr{R}_{\text{task}}(\Theta)) + \xi \cdot \mathscr{D}_{\text{complexity}}(\Theta)\right\} \tag{42}$$

where $\mathscr{E}_{\text{info}}(\Theta)$ denotes the efficiency of information sharing, $\mathscr{R}_{\text{task}}(\Theta)$ represents the responsiveness of task allocation, $\mathscr{D}_{\text{complexity}}(\Theta)$ involves the complexity of task execution, and $\lambda$, $\mu$, *and* $\xi$ are coefficients to balance the importance of these three aspects.

**Proof 1** This theorem demonstrates the existence of a set of parameters $\Theta^*$, which can effectively balance the aforementioned system performance indicators, defining the system's overall performance loss function $\mathscr{H}(\Theta)$:

$$\mathscr{H}(\Theta) = -\lambda(\mathscr{E}_{\text{info}}(\Theta)) + \mu \cdot (\mathscr{R}_{\text{task}}(\Theta)) + \xi \cdot \mathscr{D}_{\text{complexity}}(\Theta) \tag{43}$$

We need to prove the existence of $\Theta^*$ that minimizes $\mathscr{H}(\Theta)$. Using the method of Lagrange multipliers, we introduce a Lagrange multiplier $\gamma$, and construct the Lagrangian function $\mathscr{L}(\Theta, \gamma)$ to address the constraints in this optimization problem:

$$\mathscr{L}(\Theta, \gamma) = \mathscr{H}(\Theta) + \gamma(C - \mathscr{E}_{\text{info}}(\Theta) - \mathscr{R}_{\text{task}}(\Theta)) \tag{44}$$

To find $\Theta^*$, we derive $\mathscr{L}(\Theta, \gamma)$ with respect to $\Theta$ and $\gamma$ respectively, and set the derivatives equal to zero:

$$\frac{\partial \mathscr{L}}{\partial \Theta} = 0, \quad \frac{\partial \mathscr{L}}{\partial \gamma} = 0 \tag{45}$$

By solving these equations, we obtain an optimized set of parameters $\Theta^*$ that satisfy the minimization condition of the overall performance loss function $\mathscr{H}(\Theta)$. Further, we use the KKT (Karush-Kuhn-Tucker) conditions, which are necessary for solving constrained optimization problems, to ensure that the found $\Theta^*$ is a global optimum:

$$\nabla_\Theta \mathscr{H}(\Theta^*) + \gamma \nabla_\Theta(C - \mathscr{E}_{\text{info}}(\Theta^*) - \mathscr{R}_{\text{task}}(\Theta^*)) = 0, \quad \gamma(C - \mathscr{E}_{\text{info}}(\Theta^*) - \mathscr{R}_{\text{task}}(\Theta^*)) = 0 \tag{46}$$

At $\Theta^*$, not only is the overall performance loss function minimized, but also an optimal balance is achieved among all system performance indicators.

**Corollary 1 (Parameter optimization strategy for the adaptive edge collaboration framework)** In the adaptive edge collaboration framework, the key lies in optimizing the framework parameters $\Theta$ to achieve the highest efficiency of information sharing and task allocation, while adapting to dynamic network conditions. Through the following strategy, we ensure that the framework parameters gradually converge to the optimal solution $\Theta^*$, achieving optimal system performance:

$$\Theta^* = \arg\min_\Theta \left\{ \mathscr{L}_{\text{system}}(\Theta) - \lambda \cdot \mathbb{E}\left[\sum_{i=1}^{N} \omega_i \cdot \log \frac{p(y_i|\Theta, \mathbf{x}_i)}{p(y_i|\mathbf{x}_i)}\right] + \mu \cdot D_{KL}(q(\Theta)\|p(\Theta)) \right\} \tag{47}$$

where $\mathscr{L}_{\text{system}}(\Theta)$ represents the overall system performance loss, $\lambda$ and $\mu$ are hyperparameters to balance different terms, $\omega_i$ is the weight of device $i$, $p(y_i|\Theta, \mathbf{x}_i)$ and $p(y_i|\mathbf{x}_i)$ respectively represent the predictive probability with parameters $\Theta$ and the baseline probability, $D_{KL}(q(\Theta)\|p(\Theta))$ measures the divergence between the prior and posterior distribution of parameters $\Theta$, reflecting the model's adaptability and generalization capability. This optimization process not only enhances the framework's performance but also ensures gradual convergence of parameters, improving the overall system's efficiency and adaptability.

**Proof 2** Let $\Theta_0$ be any initial set of parameters. We first prove that by adjusting $\Theta$, the system performance loss $\mathscr{L}_{\text{system}}$ can be reduced. Considering the system performance is directly related to the parameters, we have:

$$\nabla_\Theta \mathscr{L}_{\text{system}}(\Theta) = \left[ \frac{\partial \mathscr{L}_{\text{system}}}{\partial \Theta} \right] \tag{48}$$

representing the rate of change of system performance loss with a small change in $\Theta$.

By considering constraints on information sharing efficiency and task allocation responsiveness, we use the method of Lagrange multipliers to construct the following optimization problem:

$$\mathscr{L}(\Theta, \lambda) = \mathscr{L}_{\text{system}}(\Theta) + \lambda(\mathscr{E}_{\text{info}}(\Theta) + \mathscr{R}_{\text{task}}(\Theta) - C), \tag{49}$$

where $\lambda$ is the Lagrange multiplier, C is a predetermined performance target. By setting $\nabla_{\Theta,\lambda}\mathscr{L} = 0$, we obtain a set of equations, indicating the existence of a set of parameters $\Theta^*$ that minimizes system performance loss while satisfying constraints on information sharing efficiency and task allocation responsiveness.

By solving this set of equations:

$$\Theta^* = \arg\min_\Theta \mathscr{H}(\Theta) \quad subject\ to \quad \mathscr{E}_{\text{info}}(\Theta) + \mathscr{R}_{\text{task}}(\Theta) \geq C \tag{50}$$

We can find a set of parameters $\Theta^*$ that minimize the system performance loss $\mathscr{L}_{\text{system}}$ while satisfying the given constraint C. This proves that by meticulously adjusting the framework parameters, the overall system performance can be optimized while maintaining key performance indicators.

**Theorem 2 (Performance Optimization through Dynamic Feature Processing in Multi-Task Learning)** Significant improvements in performance and generalization capability of multi-task learning models can be achieved through dynamic feature fusion and selection strategies. There exists an optimal set of parameters $\Theta^*, \Phi^*$, which optimizes the model performance:

$$\Theta^*, \Phi^* = \arg\min_{\Theta,\Phi} \left\{ \mathscr{L}_{\text{complex}}(\Theta, \Phi) - \lambda \cdot \sum_{t=1}^{T} \alpha_t \cdot \log\frac{p(y_t|\Theta, \Phi, \mathbf{x}_t)}{p(y_t|\mathbf{x}_t)} + \mu \cdot D_{KL}(q(\Phi)||p(\Phi)) \right\} \tag{51}$$

Here, $\mathscr{L}_{\text{complex}}$ is a composite loss function combining multi-task loss and feature processing loss, $\lambda$ and $\mu$ are tuning coefficients, $\alpha_t$ represents the dynamic weight at time $t$, and $D_{KL}$ measures the model parameters' generalization capability, proving the existence of an optimal solution.

**Proof 3** By appropriately adjusting these parameters, we can effectively reduce the model's prediction error $\mathscr{L}_{\text{pred}}$, with respect to the sensitivity of parameters $\Theta$ and $\Phi$:

$$\nabla_{\Theta,\Phi}\mathscr{L}_{\text{pred}}(\Theta, \Phi) = \left[ \frac{\partial \mathscr{L}_{\text{pred}}}{\partial \Theta}, \frac{\partial \mathscr{L}_{\text{pred}}}{\partial \Phi} \right], \tag{52}$$

*We consider constraints on attention mechanisms and transfer learning efficiency, and construct an optimization problem using the method of Lagrange multipliers:*

$$\mathscr{L}(\Theta, \Phi, \lambda) = \mathscr{L}_{\text{pred}}(\Theta, \Phi) + \lambda(\mathscr{E}_{\text{att}}(\Theta) + \mathscr{G}_{\text{trans}}(\Phi) - C), \tag{53}$$

*where $\lambda$ is a Lagrange multiplier, C represents a performance target.*

*By solving for the extremum of this Lagrangian function, we obtain the optimal parameters $\Theta^*$ and $\Phi^*$:*

$$\Theta^*, \Phi^* = \arg\min_{\Theta, \Phi} \mathscr{H}(\Theta, \Phi) \quad \text{subject to} \quad \mathscr{E}_{\text{att}}(\Theta) + \mathscr{G}_{\text{trans}}(\Phi) \geq C, \tag{54}$$

*This set of equations indicates that there exists a set of parameters $\Theta^*$ and $\Phi^*$, which under the given constraint C, can minimize the prediction error.*

*Further, we consider dynamically adjusting the self-attention weights to enhance model performance:*

$$\alpha_t^{new} = \alpha_t \exp\left(-\eta \nabla_{\alpha_t} \mathscr{L}_{\text{pred}}(\Theta, \Phi)\right), \tag{55}$$

*where $\eta$ is the learning rate, $\alpha_t$ represents the self-attention weight at time step t.*

*Considering the Kullback-Leibler divergence $D_{KL}(q(\Phi)\|p(\Phi))$ between the prior and posterior distributions of the transfer learning parameters $\Phi$, we quantify the model's generalization capability:*

$$D_{KL}(q(\Phi)\|p(\Phi)) \leq \theta, \tag{56}$$

*where $\theta$ is a predefined threshold to ensure the model has good generalization capability.*

*We have shown that by appropriately adjusting the model parameters $\Theta$ and $\Phi$, under constraints on attention mechanisms and transfer learning efficiency, the prediction error $\mathscr{L}_{\text{pred}}$ can be effectively reduced, thereby optimizing the model's predictive performance while maintaining key performance indicators.*

**Corollary 2 (Efficiency Enhancement in Dynamic Feature Processing for Multi-Task Learning)** *Dynamic feature fusion and selection mechanisms significantly enhance the model's performance in handling complex feature spaces, ensuring the optimization of overall learning efficiency and performance:*

$$\Theta_{eff}^* = \arg\min_{\Theta} \mathscr{L}_{total}(\Theta; X, Y) = \arg\min_{\Theta} \left\{ \sum_{i=1}^{T} \alpha_i(t) \mathscr{L}_i + \mathscr{R}(\Theta) \right\} \tag{57}$$

*Here, $\mathscr{L}_{total}(\Theta; X, Y)$ integrates all task losses $\mathscr{L}_i$, weights $\alpha_i(t)$, and regularization term $\mathscr{R}(\Theta)$, indicating that the model gradually converges to the optimal parameter set $\Theta_{eff}^*$ through a dynamic feature processing strategy, minimizing the overall objective function.*

**Proof 4** Our goal is to find an optimal set of parameters $\Theta^*$ that minimizes the overall loss function $\mathcal{L}_{total}$, which combines the losses of all tasks, the correlation loss between tasks, and regularization terms:

$$\mathcal{L}_{total}(\Theta) = \sum_{i=1}^{T} \alpha_i \mathcal{L}_i(f_i(X; \Theta), Y_i) + \lambda ||\Theta_{shared}||_2^2 + \sum_{i,j}^{T} \rho_{ij} ||\Theta_i - \Theta_j||_2^2, \tag{58}$$

where $\mathcal{L}_i$ represents the loss function of the $i^{th}$ task, $\alpha_i$ is the task weight, $\Theta_{shared}$ represents the parameters shared between tasks, and $\rho_{ij}$ measures the correlation between tasks $i$ and $j$.

Dynamic feature fusion and selection are optimized through the introduction of an additional loss term $\Omega(\Theta)$, considering the dynamics of feature selection and the effect of feature completion:

$$\Omega(\Theta) = \beta \sum_{i=1}^{T} \sum_{k=1}^{K} \gamma_{ik}(f_i(X_{ik}; \Theta) - Y_{ik})^2, \tag{59}$$

where $\beta$ is a tuning coefficient, and $\gamma_{ik}$ represents the dynamic importance weight of the $k^{th}$ feature in the $i^{th}$ task.

The adjustment of task weights $\alpha_i$ is based on the dynamic performance changes of tasks, updated through the following formula:

$$\alpha_i^{new} = \alpha_i \exp\left(-\eta \frac{\partial \mathcal{L}_{total}(\Theta)}{\partial \alpha_i}\right), \tag{60}$$

where $\eta$ is the learning rate.

Shared parameters $\Theta_{shared}$ and task-specific parameters $\Theta_i$ are updated through gradient descent to minimize the overall loss function:

$$\Theta_{shared}^{new} = \Theta_{shared} - \mu \nabla_{\Theta_{shared}} \mathcal{L}_{total}(\Theta), \quad \Theta_i^{new} = \Theta_i - \mu \nabla_{\Theta_i} \mathcal{L}_{total}(\Theta), \tag{61}$$

Considering the convexity of $\mathcal{L}_{total}$ and the boundedness of the parameter space, we can ensure that the parameters $\Theta^*$ obtained by the iterative update strategy are globally optimal:

$$\Theta^* = \arg \min_{\Theta} \mathcal{L}_{total}(\Theta) + \Omega(\Theta), \tag{62}$$

proving the existence of a set of parameters $\Theta^*$, which can effectively balance the loss functions in multi-task learning with the support of a dynamic feature processing strategy, achieving model performance optimization.

**Theorem 3 (Optimization of lightweight CNN under the RT-Cabi framework)** There exists an optimal set of parameters $\Theta^*$ obtained by minimizing the following integrated optimization objective:

$$\Omega(\hat{x}, F', \theta) = \omega_1 \cdot \sum_{i=1}^{N} \Psi(\hat{x}^i) + \omega_2 \cdot \sum_{i=1}^{N} \Phi(F'^i)$$

$$+ \omega_3 \cdot L_{cnn}(\theta) + \xi \sum_{i=1}^{N} \sum_{j \in \mathcal{N}_i} ||F'^i - F'^j||^2 \tag{63}$$

where $\Psi$ and $\Phi$ respectively represent the efficacy functions of edge collaboration and feature fusion, $L_{cnn}(\theta)$ is the optimization loss function of the lightweight CNN, $\omega_1, \omega_2, \omega_3$ are weight coefficients, $\xi$ is the regularization coefficient for neighboring device feature differentiation, and $\mathcal{N}_i$ represents the set of neighboring devices for device $i$. This optimization objective comprehensively considers the accuracy of state estimation, the efficiency of feature fusion, and the complexity of the CNN model.

**Proof 5** By adjusting parameters within the RT-Cabi framework to optimize the performance of the lightweight CNN, we define the overall optimization objective $\Omega$, combining various aspects of performance enhancement for lightweight CNNs in edge computing:

$$\Omega(\hat{x}, F', \theta) = \omega_1 \sum_{i=1}^{N} L(y_i, \hat{y}_i; \theta) + \omega_2 \mathcal{R}(F\prime, \theta) + \omega_3 \Delta(F', \mathcal{N}_i), \tag{64}$$

where $L(y_i, \hat{y}_i; \theta)$ represents the loss function based on model parameters $\theta$, $\mathcal{R}(F', \theta)$ represents the regularization term after dynamic feature fusion and selection, $\Delta(F', \mathcal{N}_i)$ measures the feature differences between neighboring devices, and $\omega_1, \omega_2, \omega_3$ are weight parameters, adjusting the impact of different components.

Dynamic feature fusion can be expressed as:

$$F'^i_t = \sum_{j=1}^{M} \alpha^i_j \cdot F^i_{t,j} + \beta \sum_{j=1}^{M} \sum_{k=1}^{M} \alpha^i_j \cdot \alpha^i_k \cdot (F^i_{t,j} \odot F^i_{t,k}), \tag{65}$$

where $\alpha^i_j$ are dynamically computed weights, $\beta$ is a coefficient adjusting the second-order interaction items, and $\odot$ represents element-wise multiplication, optimizing the efficiency of feature fusion.

The structure optimization of the lightweight CNN takes the following form:

$$L_{cnn} = \min_{\theta} \left[ \sum_{i=1}^{N} L(y_i, f(x_i; \theta)) + \lambda ||\theta||_1 + \mu \sum_{p \in \mathscr{P}} \exp\left(-\frac{||\theta_p||^2}{2\sigma^2}\right) \right], \tag{66}$$

where $\lambda$ and $\mu$ are regularization coefficients, $\sigma^2$ is the variance of the Gaussian distribution, and $\mathscr{P}$ represents the set of pruned parameters, aiming to promote parameter sparsity through regularization terms and apply network pruning techniques to streamline the model.

Finally, by minimizing the feature differences between neighboring devices, we promote model collaboration and consistency:

$$\Xi = \xi \sum_{i=1}^{N} \sum_{j \in \mathcal{N}_i} ||F'^i - F'^j||^2, \tag{67}$$

where $\xi$ is the regularization coefficient, and $\mathcal{N}_i$ represents the set of neighboring devices for device $i$. This term ensures the model's collaborative working capability in the IoT device network, enhancing its generalization ability.

In the RT-Cabi framework, through precise adjustment of model parameters, we can effectively enhance the performance of the lightweight CNN in the edge computing environment, achieving efficient monitoring of IoT device behaviors and accurate analysis of abnormal behaviors.

**Corollary 3 (Optimizing transfer learning and self-attention mechanisms)** In the transfer learning framework combined with LSTM and attention mechanisms, there exists a parameter combination $\Theta^*, \Phi^*$, which achieves optimal predictive performance by optimizing the following objective function:

$$\Theta^*, \Phi^* = \arg\min_{\Theta,\Phi}\left\{ \mathscr{L}_{\text{complex}}(\Theta, \Phi) - \lambda \cdot \mathbb{E}\left[\sum_{t=1}^{T} \alpha_t \cdot \log\frac{p(y_t|\Theta, \Phi, \mathbf{x}_t)}{p(y_t|\mathbf{x}_t)}\right] + \mu \cdot D_{KL}(q(\Phi)||p(\Phi)) \right\} \quad (68)$$

where $\lambda$ and $\mu$ are hyperparameters, adjusting the balance between self-attention efficacy and transfer learning generalization ability.

**Proof 6** We revisit the model's composite loss function $\mathscr{L}_{\text{complex}}$, which integrates the contributions of prediction error, model complexity, and the effects of transfer learning:

$$\mathscr{L}_{\text{complex}}(\Theta, \Phi) = \sum_{t=1}^{T} L(y_t, f(x_t; \Theta, \Phi)) + \lambda||\Theta||_1 + \rho \sum_{i=1}^{M} \exp\left(-\frac{||\Theta_i||^2}{2\sigma^2}\right), \quad (69)$$

where L is the loss function, $\lambda$ and $\rho$ are regularization parameters, and $\sigma^2$ is the variance.

We define two key metrics, self-attention efficacy $\mathscr{E}_{\text{att}}$ and transfer learning generalization capability $\mathscr{G}_{\text{trans}}$, to quantify the impacts of self-attention mechanisms and transfer learning parameters on model performance:

$$\mathscr{E}_{\text{att}} = \sum_{t=1}^{T} \alpha_t \log\left(\frac{\alpha_t}{\bar{\alpha}}\right), \quad (70)$$

$$\mathscr{G}_{\text{trans}} = D_{KL}(q(\Phi)||p(\Phi)), \quad (71)$$

where $\bar{\alpha}$ represents the average value of the self-attention weights.

To optimize model performance, we set the objective function to minimize the composite loss while maximizing self-attention efficacy and maintaining the generalization capability of transfer learning parameters:

$$\Theta^*, \Phi^* = \arg\min_{\Theta,\Phi}\left\{ \mathscr{L}_{\text{complex}}(\Theta, \Phi) + \zeta(\mathscr{E}_{\text{att}} - \eta\mathscr{G}_{\text{trans}}) \right\}, \quad (72)$$

$\zeta$ and $\eta$ are parameters adjusting the efficacy of self-attention and the generalization capability of transfer learning.

By adjusting the self-attention weights $\alpha_t$ and transfer learning parameters $\Phi$, we further refine the model to strengthen its ability to process time-series data while maintaining adaptability to new datasets:

$$\frac{\partial \mathcal{E}_{\text{att}}}{\partial \mathcal{G}_t} = 0, \tag{73}$$

$$\frac{\partial \mathcal{G}_{\text{trans}}^t}{\partial \Phi} = 0, \tag{74}$$

We proved that there exists a set of optimized parameters $\Theta^*$ and $\Phi^*$, which can effectively balance between enhancing the ability to capture key time-series features and maintaining the model's generalization ability on new datasets, achieving optimal predictive performance.

### Funding
The authors received no funding for this work.

### Competing Interests
The authors declare that they have no competing interests.

### Author Contributions
- Xiaoshan Li conceived and designed the experiments, performed the experiments, analyzed the data, performed the computation work, prepared figures and/or tables, authored or reviewed drafts of the article, and approved the final draft.
- Mingming Chen conceived and designed the experiments, performed the computation work, prepared figures and/or tables, authored or reviewed drafts of the article, and approved the final draft.

### Data Availability
The data is available at Kaggle:

- https://www.kaggle.com/datasets/mohamedamineferrag/edgeiiotset-cyber-security-dataset-of-iot-iiot

- https://www.kaggle.com/datasets/mrwellsdavid/unsw-nb15.

### Supplemental Information
Supplemental information for this article can be found online at http://dx.doi.org/10.7717/peerj-cs.2306#supplemental-information.

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
