# Peer review of "RT-Cabi: an Internet of Things based framework for anomaly behavior detection with data correction through edge collaboration and dynamic feature fusion"

_PeerJ Computer Science, doi:10.7717/peerj-cs.2306_

## Round 0.1 · original submission · Major Revisions

Please follow the reviewer comments in your revision

Reviewer 1 ·

Basic reporting

See "Additional Comments"

Experimental design

See "Additional Comments"

Validity of the findings

See "Additional Comments"

Additional comments

The paper entitled “RT-Cabi: An IoT-based Framework for Anomaly Behavior Detection with Data Correction Through Edge Collaboration and Dynamic Feature Fusion” introduces RT-Cabi, a framework for anomaly behavior detection in IoT traffic using edge computing. RT-Cabi leverages edge computing to enhance data processing and analysis capabilities, improving accuracy and efficiency. The framework incorporates an adaptive edge collaboration mechanism, dynamic feature fusion and selection techniques, and optimized lightweight convolutional neural network (CNN) tools. The authors claim that RT-Cabi achieves high detection accuracy on two public datasets, Edge-IIoT and UNSW NB15, outperforming existing methods. The manuscript also includes an experimental part.

Strengths:

* The paper addresses an important problem in IoT security (anomaly behavior detection in traffic flow).
* The proposed framework leverages edge computing to enhance the data processing and analysis capabilities to improve the accuracy and efficiency of anomaly detection.
* The framework incorporates an adaptive edge collaboration mechanism, dynamic feature fusion and selection techniques, and optimized lightweight CNN frameworks.


This is a list of the main points to address:
* The manuscript is not simple to read and follow, especially the chapter 2. Some sentences are too long, and the mathematical part is not easy to follow. Try to use simplify this part and reduce the number of pages, increasing the readability.
* The paper lacks a clear description of the experimental setup, making it difficult to replicate the results. Moreover, the paper does not provide a rigorous comparison with existing methods, making it difficult to assess the performance of RT-Cabi.
* The authors limit their evaluation to class performance measures (e.g., accuracy), but seem to neglect the temporal complexity associated with their framework. Specifically, regarding feature selection, this stage is crucial for making the prediction more flexible. However, it is also time-consuming, and some attacks could be performed before the feature selection/extraction task is completed. Therefore, the authors are encouraged to discuss this problem and refer to recent literature on this topic. Some suggestions include:
- "Network Intrusion Detection System using Feature Extraction based on Deep Sparse Autoencoder", (IEEE ICTC conference, 2020);
- "Multi-Stage Optimized Machine Learning Framework for Network Intrusion Detection", (IEEE ,Trans. on Netw. and Serv. Management, 2020);
- "Supervised Feature Selection Techniques in Network Intrusion Detection: a Critical Review",(Elsevier, Engineering applications of artificial intelligence, 2020).

Overall, the paper presents an interesting framework for anomaly behavior detection in IoT traffic using edge computing. However, the lack of a clear description of the experimental setup and the limited comparison with existing methods make it difficult to assess the performance of RT-Cabi. The authors should address these issues to improve the clarity and rigor of the paper.

Reviewer 2 ·

Basic reporting

The authors propose a new "edge computing" friendly framework named RT-Cabi for anomaly detection within the IoT. The paper is overall written in a clear and professional English. The authors provide a detailed introduction, sufficient context and background, detailed information about the framework and algorithms used. Formal results are supported by relevant proofs and theorems.

I would suggest the following improvements to increase the quality of the manuscript:

- Text at 35-37 could be made clearer, as currently it is difficult to understand the trend the authors refer to and the context; also, at row 16 I would suggest adding a year or date by which the increase is supposed to happen in order to clarify and make it clear for the reader also in the future;
- The sentence at 38-39 could benefit from rephrasing in order to increase clarity;
- For the literature review sections, I would suggest a short check for maybe new relevant related work - particulary section 1.2.2 where the most recent referenced work seems to be from 2020;
- In section 1.3, rows 137-138: I would suggest that the authors expand the text there in order to better highlight and recap how the proposed framework mitigates or solves the current gaps in the state-of-the-art referenced in previous sections. Furthermore, I believe that the authors should explain also via a pargraph Figure 1. Also, the bullet point list lacks an introduction.
- row 380 - 10000 samples out of how many? Presented in this way the reader cannot assess if this number is large, small, appropriate considering the dataset.
- row 381-382: I would suggest that further details are being given about the type of corrections made. It is currently not clear what type of corrections were made and how the missing data issue was mitigated.
- row 384: it is unclear to me what the meaning of "deployed within 2 hours" is. I would suggest rephrasing in order to improve clarity.
- row 387: It is not mentioned throughout the manuscript the type of resource contrained devices that the authors target. I believe it would be useful to include some examples and expand a bit on the suitability of the approach for such a chosen IoT device.
- row 394: I could not find an explanation throughout the manuscript of what a traditional CNN means for the authors. I would suggest adding further details to make sure that both the reader and the authors are thinking of the same thing.
- Table 3. I would suggest an improvement on the reporting of the time cost and space cost - e.g. model deployment time of 2 hours does not really give much information - what is meant by it - what parameteres were considered - how can different parameters influence it (e.g. number of epochs, hardware etc.)
- Figure 4 and Figure 5 could benefit from more detailed captions. Also, an explanation about how the value of 20 rounds has been chosen should be given somewhere in the text.
- In section 4.4 the table information does not align with the text - e.g. RT-Cabi's performance for the UNSW-NB15 dataset is different in the text compared to the table. I would suggest further checking and corrections. Also, are there further relevant methods from the related work sections that RT-Cabi could be compared against?
- In section 5, I would suggest that the authors expand the current text with highlighting also possible limitations or future work directions for their work.

Experimental design

The experimental design seems to be sound and the code provided does encourage replicability and further investigation by interested parties. Furthermore, there is information also within the manuscript describing the algorithms and the theorethical concepts employed.

Validity of the findings

The finding seem to be relevant and promising. Perhaps the authors can include more details about an edge computing set-up where their framework could be deployed - in terms of computational capability, examples of IoT devices etc. Also future work directions and promising avenues should be explicitly given as to encourage further contributions and use of the provided code by interested parties.

---

## Round 0.2 · accepted · Accept

I confirm the revisions are done properly

Reviewer 1 ·

Basic reporting

In this revised version, the authors made a good effort to address all my comments raised in the previous round of review. In particular, the authors have:

- Simplified some parts of the paper;
- Elaborated more in detail the experimental part through the addition of more setup parameters;
- Elaborated more in detail the time complexity problems associated to the feature selection by also comparing some recent and credited literature on the topic;

In my opinion, the paper can be now accepted in its current form.

Experimental design

N/A

Validity of the findings

N/A